# The Role of Pre- and Post-Transplant Hydration Status in Kidney Graft Recovery and One-Year Function

**DOI:** 10.3390/medicina59111931

**Published:** 2023-11-01

**Authors:** Andrejus Bura, Vaiva Kaupe, Justina Karpaviciute, Asta Stankuviene, Kestutis Vaiciunas, Inga Arune Bumblyte, Ruta Vaiciuniene

**Affiliations:** 1Department of Nephrology, Medical Academy, Lithuanian University of Health Sciences, 44307 Kaunas, Lithuania; 2Department of Urology, Medical Academy, Lithuanian University of Health Sciences, 44307 Kaunas, Lithuania

**Keywords:** kidney transplantation, graft function, hydration status, bioimpedance analysis, lung sonography

## Abstract

*Background and Objectives*: Early improvements to graft function are crucial for good outcomes in kidney transplantation (kTx). Various factors can influence early graft function. This study aimed to evaluate the pre- and post-transplant hydration statuses of kTx recipients using bioimpedance analysis (BIA) and lung ultrasonography (LUS) and to investigate the hydration status’ relationship with the function of the transplanted kidney during the first year after transplantation. *Materials and Methods*: This observational prospective cohort study included deceased kidney recipients transplanted in the Hospital of the Lithuanian University of Health Sciences between September 2016 and January 2023. BIA and LUS were performed before transplantation, on days 3 and 7, and at discharge. Data on recipient and donor clinical characteristics were collected. Graft function was evaluated according to the serum creatinine reduction ratio and the need for dialysis. Hydration status was evaluated by calculating B-lines (BL) on LUS and the ratio of extracellular/total body water on BIA. *Results*: Ninety-eight kTx recipients were included in the study. Patients with immediate graft function (IGF) were compared to those with slow or delayed graft function (SGF + DGF). Recipients in the SGF + DGF group had a higher sum of BL on LUS before transplantation. After transplantation in early postoperative follow-up, both groups showed hyperhydration as determined by BIA and LUS. After one year, recipients with no BL before transplantation had better graft function than those with BL. Logistic regression analysis showed that having more than one BL in LUS was associated with a 2.5 times higher risk of SGF or DGF after transplantation. *Conclusions*: This study found that lung congestion detected by LUS before kTx was associated with slower graft recovery and worse kidney function after 1 year. Meanwhile, the hyperhydration status detected by BIA analysis did not correlate with the function of the transplanted kidney.

## 1. Introduction

Kidney transplantation (kTx) is a treatment option for patients with end-stage renal disease (ESRD) [1]. The early improvement in graft function is important for patient outcomes. The function of a transplanted kidney depends on the cold ischemia time, expanded donor criteria, donor kidney preservation technique, human leukocyte antigen compatibility with the recipient and other factors [2]. Recipient characteristics such as type and duration of dialysis, comorbidities and age also influence early graft function [3]. Perioperative fluid prescription is important for initial graft function. Many articles have focused on the effects of fluid management and monitoring in the intraoperative and postoperative periods on the early function of transplanted kidneys. Van Loo et al. [4] showed that in hemodialysis (HD) patients, ultrafiltration in the 24-h period before transplantation increased the risk of delayed graft function (DGF). Fluid loading during the first 72 h was associated with a lower incidence of DGF, independent of central venous pressure [5]. Furthermore, the hydration status before transplantation also has to be considered.

The evaluation of the hydration status of patients with ESRD is challenging. Several articles advise on how to monitor the hydration status of patients with ESRD using different physical and instrumental methods [6,7,8,9]. The most popular objective instrumental methods are bioimpedance analysis (BIA) and lung ultrasonography (LUS). BIA is based on an alternating current to the body and measuring changes in impedance/resistance. It can be used to quantify total body water (TBW), intracellular water (ICW), extracellular water (ECW), protein levels and fat levels [8]. LUS can detect extravascular lung water before the onset of symptoms [10]. The rationale behind this technique is that in the presence of lung congestion, the ultrasound beam is reflected by thickened interlobular septa, generating hyperechoic reverberation artifacts between edematous septa and the overlying pleura (i.e., ultrasound bundles at a narrow basis going from the ultrasound transducer to the limit of the screen, the so-called lung comets, which can be considered an ultrasound equivalent of B-lines (BL) detected in chest X-rays) [10].

To our knowledge, only a few studies have attempted to assess the hydration status of kTx recipients using BIA [6,7] and no study has used LUS. Andreia Coroas et al. [6] demonstrated in their study that patients with ESRD exhibit altered water distribution, specifically an increase in total body water and extracellular water. In recipients with well-functioning kidney transplants, the different body water compartments quickly equalize to match the composition of normal individuals. Gueuten V. et al. [7] demonstrated that a lower hydration status, as determined by BIA, is associated with DGF. Our study was performed to assess the hydration status of renal transplant recipients before transplantation using BIA and LUS and to evaluate changes in hydration in the early phase after kTx. We also aimed to determine whether BIA or LUS has prognostic value for graft function in the early phase after kTx.

## 2. Materials and Methods

### 2.1. Study Design

The observational prospective cohort study included all consecutive recipients of the deceased kidney transplantation program at the Lithuanian University of Health Sciences Kauno Clinics hospital between September 2016 and January 2023. Approval (29 December 2015 Nr. P1-BE-2-9/2014) to perform this study was given by the Kauno region biomedical ethics committee. Informed consent was signed by every participant before kTx.

### 2.2. Evaluation of Early Graft Function

Twenty-two definitions of DGF have been proposed in the literature based on the need for HD, serum creatinine levels, urine output or a combination of these [11,12,13,14]. The most used definition is the requirement for HD within the first week after transplantation. However, this definition has limitations: misclassification may occur if a single HD session is needed within the first hours after transplantation due to hyperkalemia or hyperhydration [15]. In our study evaluating early graft function, we used the definition proposed by Isaac E. Hall and co-authors [16]—immediate (IGF), slow (SGF) and DGF. The authors’ suggested formula to calculate serum creatinine (Scr) reduction—the difference between Scr at 0 h and Scr on day 7 divided by Scr at 0 h—was used to perform the calculations. In those who did not require HD, SGF was defined as an Scr reduction ratio of less than 0.7, and IGF was defined as a ratio greater than or equal to 0.7. DGF was defined by at least one HD session within 7 days of transplant, excluding patients who underwent one HD session due to early postoperative hyperkalemia. Primary graft nonfunction (PNF) was defined as a permanent lack of graft function from the time of transplantation. There were 51 patients with IGF, 32 patients with SGF, 15 patients with DGF and 4 patients with PNF. Due to the small percentage of patients in the DGF, we merged them with the SGF group for the final analysis, so we compared two patient groups: IGF and SGF+DGF. PNF patients were excluded from the study.

### 2.3. Study Population

There were 120 cadaveric kTx performed during the study period. KTx recipients who received a kidney from a living donor (n = 4), had early surgical complications, renal artery or vein thrombosis (n = 4), PNF (n = 4), data limitations (n = 7) or incomplete follow-up (n = 3) were excluded. In the initial analysis, we included 98 patients. For the follow-up analysis after 1 year, we tracked 74 kidney transplant recipients out of the 98 patients (24 recipients were excluded due to data limitations after 1 year due to further patient follow-up in another transplant center). Recipients were either pre-emptive or on chronic HD or peritoneal dialysis treatment before kTx. HD immediately before kTx was performed when the patient was dialyzed more than 2 days before transplantation, had serum potassium levels > 5.5 mmol/L and/or II or III degrees of venous stasis was found in the chest X-ray. A standard immunosuppression induction was administered before kTx: methylprednisolone, mycophenolate mofetil and Basilixumab or ATG in the case of high immunological risk. The prescription of fluid therapy after kidney transplantation was determined according to the local protocol, which included a basic volume load of 1 L of isotonic solution and 1 L of 5% glucose solution with additional compensation of extra losses. All deceased kidneys were preserved using the cold storage immersion method. We collected data about recipients’ demographic and clinical characteristics, donors’ clinical characteristics and graft function in the early post-transplant phase and during the first year after transplantation. 

### 2.4. Determining the Volume Status 

For an evaluation of hydration status, we performed BIA using an InBody S10 analyzer (Biospace, Seoul, Republic of Korea), and for pulmonary congestion, LUS using a Mindray model Z5 with curvilinear probe model 35C50EA for enrolled subjects.

All subjects underwent BIA and LU from the same investigator before kTx and after kTx on days 3 and 7 and on discharge. The following variables before kTx were also included: residual diuresis, serum sodium levels and total protein levels. During the hospitalization period, data about fluid therapy on the day of transplantation and after transplantation and diuresis volume were collected.

When performing BIA, we took into account the ratio of ECW/TBW. BIA was performed according to the instructions of the manufacturer. If HD was undertaken before transplantation, BIA was performed after the HD session. 

For the methodology of LUS, we used recommendations for point-of-care lung ultrasound [17]. We used a complete eight-zone LU to evaluate the sum of BL. According to the results, we divided patients into two groups: those with a sum of 1 or more BL and those with no BL before kTx.

### 2.5. Post Hoc Analysis of Relation between Left Ventricular Hypertrophy and Overhydration

We performed a post hoc analysis on 73 patients’ echocardiography data before kTx that were available in a local database. We collected data about left ventricular mass index (LVMI) and ejection fraction. We divided patients into two groups according to left ventricular hypertrophy (LVh); according to the American Society of Echocardiography and/European Association of Cardiovascular Imaging, LVH is defined as an LVMI greater than 95 g/m in women and greater than 115 g/m in men [18]. We compared the hydration status in patients with and without LVh. 

### 2.6. Statistical Analysis

Data were analyzed using SPSS 28 statistical software. Values were expressed as means (SD) or median (IQR) according to their distribution for continuous variables or as counts and percentages for categorical variables. Categorical data were analyzed using Pearson’s chi-square exact test. The mean square contingency coefficient was used as a measure of association between two binary variables. For normally distributed continuous variables, Student’s unpaired *t*-test was used. The Kruskal–Wallis or Mann–Whitney U-test was applied if variables were not normally distributed. Logistic regression analysis was performed using binary logistic regression. All reported *p* values were two-sided. A *p* value of 0.05 or less was considered significant.

## 3. Results

### 3.1. Patient Characteristics and Clinical Data before Transplantation

A total of 98 patients were enrolled in the study between September 2016 and January 2023. Of these 98 patients, 51 recipients had IGF and 47 had SGF or DGF. A comparison of donor and recipient characteristics in two groups is shown in Table 1. Recipients from the SGF+DGF group had a higher sum of BL before kTx. This group of patients received older donors’ kidneys with a longer cold ischemic time. Intraoperative fluid administration showed no significant differences between the groups (33.22 (11.75) mL/kg vs. 33.05 (14.61) mL/kg, *p* = 0.952). The ECW/TBW ratio did not differ between the groups. 

In the unadjusted analysis, more than 1 BL in LUS was associated with a higher risk of getting SGF or DGF after kTx (OR 2.514, Cl (1.079–5.858) (model 1). However, after adjusting for potentially relevant confounders with high significance in the univariate analysis, such as cold ischemic time and deceased donor age, this relationship was no longer apparent (OR 1.169, Cl (0.941–1.452) (model 2). (Table 2). Given that there were only 47 events with SGF and DGF, there is always the potential for an overestimation of the association strengths when using multivariate analysis. However, a compelling argument in favor of B lines is that model 2 (Table 2) without BL resulted in an increased Akaike information criterion (AIC) and a decreased Nagelkerke R2 compared to when BL was included: 130.910 and 0.114 versus 122.47 and 0.2, respectively.

### 3.2. Postoperative Data on Days 3 and 7 and on Discharge

Comparing the groups of patients with IGF and SGF+DGF, the hydration status was not different between BIA and LUS on day 3 after transplantation. Fluid therapy per 24 h on day 3 was administered at an equal volume to the groups (46.86 (17.68) mL/kg/24 h vs. 39.55 (17.42) mL/kg/24 h, *p* = 0.068). Early infection status was excluded in all patients; however, in the group SGF+DGF, the serum CRP level was significantly higher (17.2 (7.5–26.0) mg/L vs. 27.0 (11.9–46.56) mg/L, *p* = 0.013). In the SGF+DGF group, the measured glomerular filtration rate (mGFR) and 24-h diuresis were significantly lower compared to patients with IGF. More details are shown in Table 3.

On the seventh day after kTx, all patients were in a hyperhydration state as determined by BIA and LUS methods. The tendency for lung congestion to reduce faster was observed in the IGF group (3.2 (3.39) vs. 4.83 (4.55), *p* = 0.064); however, the volume of fluid therapy per 24 h was significantly higher in this group (54.68 (18.13) mL/kg/24 h vs. 38.83 (34.58) mL/kg/24 h) (Table 4). There was a positive Spearman correlation between BL and ECW/TBW at day 7 post-transplantation (r 0.356, *p* = 0.004), as shown in Figure 1.

On discharge day, both groups of recipients underwent a normohydration state in the lungs, as determined by LUS; however, the BIA analysis showed a hyperhydration state in both groups of patients. The kTx function was good in both groups; however, mGFR was lower in the SGF + DGF group (64.5 (24.23) mL/kg/1.73 m^2^ vs. 41.57 (17.79) mL/kg/1.73 m^2^, *p* = 0.001) and patients in this group were more anemic (Table 5). 

### 3.3. Changes in Volemic Status during the Observation Period

The manufacturer of the BIA analysis equipment provides a cut-off point of >0.390 for ECW/TBW to state overhydration. In both groups, overhydration was not observed before kTx and overhydration was established in both groups during the observation period after kTx without significant differences between the groups. The BL sum showed a hyperhydration state of the lungs before kTx in the SGF + DGF group (0.92 (1.67) vs. 2.07 (3.25), *p* = 0.037), and during observation periods, lung congestion was not reduced until discharge day, as shown in Figure 2.

During the follow-up period after kTx, all participants were found to have excess hydration, which was determined by the two methods LUS and BIA, as shown in Figure 3a,b.

The Pearson chi-square test showed that patients without BL in LUS before kTx were related to IGF (33 pt vs. 21 pt, *p* = 0.031) with a positive relationship Phi correlation of 0.224, *p* = 0.031. On days 3 and 7 and discharge days, no relation was found between BL and the Scr reduction ratio. No correlation was found between the ECW/TBW ratio before and after the kTx and Scr reduction ratio.

### 3.4. Relation of the Recipient’s Volemic Status and Graft Function after 1 Year

During the 1-year follow-up period, 74 out of 98 patients were observed. The mean function of the transplanted kidney after 1 year was 46.85 (SD 20.77) mL/min/1.73 m^2^, as determined by the CKD-EPI equation. The function of the transplanted kidney was better in recipients with no BL before transplantation (n = 29) compared to 1 or more BL before kTx (n = 45) (52.09 (SD 19.89) vs. 38.9 (SD 17.73) mL/min/1.73 m^2^ by CKD-EPI equation, *p* = 0.034). 

A moderate negative correlation was found between the sum of BL before transplantation and the estimated glomerular filtration rate (eGFR) after 1 year (−0.256, *p* = 0.033).

For the analysis of data of BIA, patients were divided into hyperhydration (ECW/TBW > 0.390) and no hyperhydration (ECW/TBW ≤ 0.39) groups before kTx. No significant differences were found between the groups in terms of eGFR and Scr levels at 1-year follow-up. Additionally, no correlation was found between ECW/TBW ratio and eGFR after 1 year.

To evaluate the importance of BL in LUS in multivariate analysis, the cohort was divided into two groups: those with good kidney function after 1 year (eGFR ≥ 30 mL/min/1.73 m^2^) and those with worse kidney function after 1 year (eGFR < 30 mL/min/1.73 m^2^). In the group with worse kidney function, there were 13 patients, accounting for 17.6% of the total. An adjusted analysis was performed, taking into account relevant cofounders: more than 1 BL in LUS, the age of the deceased donor and cold ischemic time. However, only the age of the donor remained a significant cofounder (Table 6).

### 3.5. Post Hoc Analysis of Relation between Left Ventricular Hypertrophy and Overhydration

In the LVh group, there were 49 patients (67%). The sum of BL before kTx (1 (SD 1.56) vs. 1.8 (SD 3.74), *p* = 0.067) and ECW/TBW ratio (0.378 (0.373–0.383) vs. 0.384 (0.368–0.389), *p* = 0.142) were not different in the groups with and without LVh. We did not find a correlation between the LVMI and the sum of BL and ECW/TBW before kTx. The ejection fraction did not show a significant difference between groups, and no correlation between ejection fraction and BL, or ECW/TBW before kTx, was observed.

## 4. Discussion

This study focused on changes in the patient’s hydration state after kTx and its relation to graft function. We hypothesized that a hyperhydrated state would be beneficial for early and later kidney function but our study showed the opposite: signs of lung congestion before kTx were associated with SGF recovery and worse kidney function after 1 year. This association was found using LUS but not BIA analysis for evaluating the hydration state. This study suggests a new possibility of using LUS before kTx to predict the early function of the transplanted kidney and function after 1 year and to make early interventions.

Similar results were presented by Smudal A. et al. According to their study, DGF was related to lower residual diuresis, higher intradialytic weight gain and higher fluid balance during the first postoperative day [19]. Another study shows that a hypervolemic state measured by central venous pressure before kTx is associated with a greater risk of chronic dysfunction. The author mentions possible reasons, such as the deleterious effects of fluid overload on cardiovascular and pulmonary physiology, which produce impaired cardiac output, possibly leading to worse outcomes [20]. Germain M.J. compared 57 patients’ estimated dry weight before kTx and dry weight 2 weeks after kTx. The authors found that achieving the correct estimated dry weight during the HD stage is challenging; in the study, 75% of patients who underwent kTx had an inaccurately estimated dry weight before kTx [21].

The hydration status of the patient can be determined with various principles. We chose noninvasive methods: BIA and LUS. BIA analysis shows the whole-body hydration status and LUS shows the lungs’ extracellular water in the lung parenchyma. Both methods are cheap, harmless and quick and can be performed close to the patient’s bedside. Our study shows that all patients underwent a hypervolemic state after kTx, as determined by both methods. However, we did not find a correlation between these methods. According to Panuccio V.’s pilot study, LUS showed more accurate results in detecting lung congestion at a preclinical stage in euvolemic acute kidney injury patients [22]. In contrast to our study, a correlation between the sum of BL and dry weight assessed by BIA was found by Vitturi N et al. [23]. Therefore, we can conclude that LUS and BIA are two overlapping methods used to determine the hydration status.

To avoid the incidence of DGF, high-volume fluid is administered, aiming to preserve renal blood flow. Consequently, we could achieve a hypervolemic state. We observed that all patients underwent a hyperhydration state after kTx, as assumed by BIA and LUS. The same results were found in Guetin V.’s study. The authors showed that patients after kTx are associated with hydration disorders, established by BIA, and, like in our study, all patients after kTx were in a hyperhydration state [7].

The question is what volume of fluids is not harmful or critical to the patient. Volume overload can cause respiratory complications, infection and increased overall mortality [9,24,25]. Furthermore, systemic venous congestion and volume overload can decrease the perfusion of the kidney and prolong the ischemic period [25]. In our study, we determined the possible consequences of hypervolemia with signs of lung congestion before kTx, as these patients underwent SGF and DGF early after kTx. We could hypothesize that knowing the hydration status of these patients before kTx could prevent fluid therapy overdosage. According to Eduardo R Argaiz et al., the presence of BL on LUS constitutes a strong risk indicator for fluid administration, regardless of whether it has a cardiogenic origin or not [25].

The lungs are a critical area at the interface with the central circulation [26]. Studies have shown that the number of BL in HD patients reflects LV filling pressure, which is a fundamental metric of central circulation [27]. Loutradis Ch. found that the reduction in dry weight, according to the sum of BL in follow-up, changed sizes in the left and right atrium, LV and LV filling [28]. In our study, no difference was found in the hydration status of patients with and without LVh, as determined by the two methods before kTx.

Fluid overload, intermittent HD and uremic intoxication are risk factors for damaging endothelial glycocalyx in patients undergoing HD [29,30]. The consequences of glycocalyx perturbation include a wide range of vascular abnormalities in experimental models, including increased vascular permeability followed by the generation of tissue edema [31]. Extended glycocalyx damage can also be detected in human kidney grafts during kTx (ischemic-reperfusion injury), causing vascular permeability, interstitial edema and endothelial cell swelling. Glycocalyx degradation inversely correlates with creatinine clearance [32].

We suggest that venous congestion and probably glycocalyx damage may be the reasons for our observation that fluid overload and lung congestion before transplantation are harmful to renal transplant recovery and function after transplantation. LUS data may show us the consequences of highly damaged glycocalyx levels in chronically hyperhydrated patients before kTx.

This study has several limitations. The main limitation is the small sample size. Although the significant hypervolemic state, as indicated by BL in LUS, was not a significant factor in the multivariate analysis, we hypothesize that, with a larger sample size, it could potentially become significant. In future studies, it will be necessary to broaden the scope of the patient hydration status assessment. Echocardiography performed before and after kTx, along with the collection of endothelial injury biomarkers or N-terminal pro B-type natriuretic peptide (NT-proBNP) samples, could provide valuable information. It is important to note that the evaluation of lung congestion does not provide a complete picture of congestion levels in the venous system. However, such an evaluation could offer additional insights into fluid intolerance and the potential for worsening systemic venous congestion and damage to the endothelial glycocalyx [25].

This is the first clinical study to investigate the value of LUS in relation to kidney transplant function during the early postoperative period.

## 5. Conclusions

This study found that LUS-revealed lung congestion before kTx was associated with slower graft recovery and worse kidney function after 1 year. Meanwhile, the hyperhydration status detected using BIA did not correlate with the function of the transplanted kidney.

## Figures and Tables

**Figure 1 medicina-59-01931-f001:**
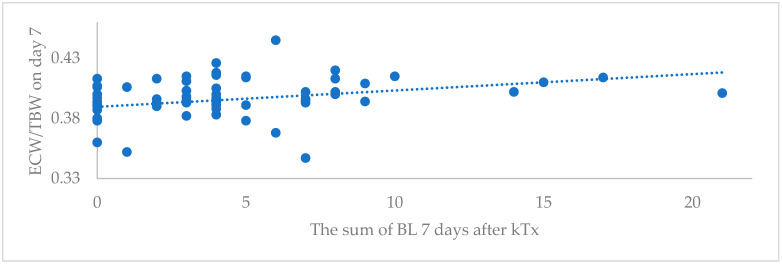
Correlation between the sum of BL and ECW/TBW on day 7 after kidney transplantation.

**Figure 2 medicina-59-01931-f002:**
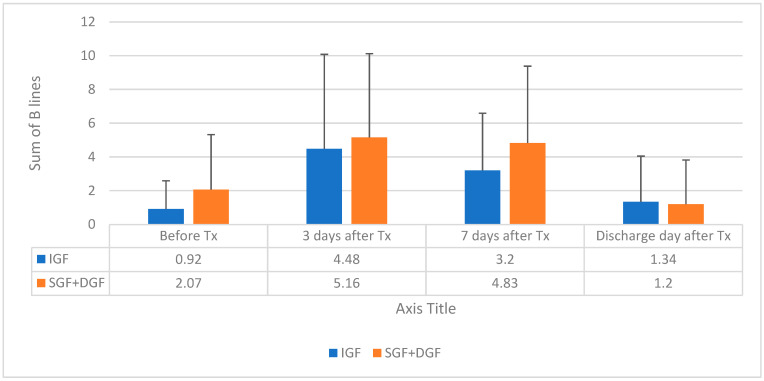
Relationship between BL and the recovery of the function of the transplanted kidney during the observation period, mean with SD.

**Figure 3 medicina-59-01931-f003:**
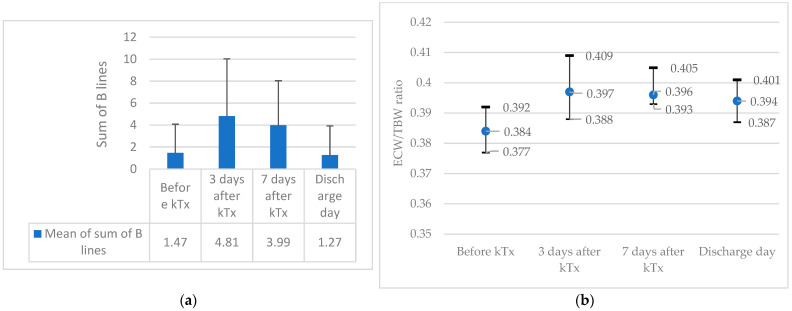
Changes in the volemic status of all kidney transplant recipients measured by LUS (**a**) (mean with SD) and BIA (**b**) (median with IQR).

**Table 1 medicina-59-01931-t001:** Relationship between demographic and clinical data before kidney transplantation and kidney function after transplantation.

	IGF ^1^	SGF + DGF ^1^	
**Recipients**	n = 51	n = 47	*p*
**Gender male**	26 (51%)	25 (53.2%)	0.827
**Age (years)**	50.65 (12.07)	47.94 (12.57)	0.279
**Duration of kidney replacement therapy (months)**	22.00 (9.0–49.0)	20.00 (9.0–37.0)	0.326
**Residual diuresis (mL/24 h)**	500 (0.0–1000.0)	600 (0.0–1500.0)	0.413
**Creatinine (μmol/L)**	774 (618.0–960.0)	646 (526.0–839.0)	0.032
**Urea (mmol/L)**	19.4 (13.2–24.7)	17.2 (11.80–23.30)	0.413
**Sodium (mmol/L)**	136 (134.0–138.0)	137 (134.0–139.0)	0.609
**Serum total protein (g/L)**	72.65 (6.51)	73.70 (5.96)	0.414
**Hemoglobin (g/L)**	123 (114–129)	123 (114.0–130.0)	0.613
**CRP (mg/L)**	5 (1.5–5.0)	3 (1.32–5.0)	0.305
**BMI (kg/m2)**	25.18 (4.69)	26.27 (5.75)	0.349
**Sum of BL ^2^**	0.92 (1.67)	2.07 (3.25)	0.037
**ECW/TBW ratio ^3^**	0.381 (0.375–0.389)	0.381 (0.374–0.390)	0.646
**Amount of fluid** **per transplantation day (mL/kg)**	33.22 (11.75)	33.05 (14.61)	0.952
**Donor data**			
**Age (years)**	46 (32.0–56.0)	54 (48.0–60.0)	0.003
**Expanded criteria donor ^4^**	5 (20%)	6 (25%)	0.347
**Diuresis per hour (mL/h)**	112.5 (87.5–166.67)	116.67 (95.83–165.0)	0.91
**Cold ischemic time of transplanted kidney (min)**	840 (720.0–1020.0)	960 (800.0–1080.0)	0.044

^1^ Determination of IGF, SGF and DGF was performed using the following formula: the difference between serum creatinine (Scr) at 0 h and Scr on day 7 divided by Scr at 0 h. In those who did not require HD, SGF was defined as a creatinine reduction ratio less than 0.7 and IGF was defined as a ratio greater than or equal to 0.7. DGF was defined by at least one HD session within 7 days of transplant. ^2^ Sum of B-lines. ^3^ ECW/TBW ratio—extracellular water/total body water ratio. ^4^ Expanded criteria donor—60 years old and more or more than 50 years old with two criteria: arterial hypertension, serum creatinine > 130 μmol/L, death underwent cerebral vascular damage. Data given as a number (%), mean (SD) or median (IQR).

**Table 2 medicina-59-01931-t002:** Multivariate logistic regression for the evaluation of factors relevant to slow and delayed graft function.

Model	Odds Ratio (95% CI)	*p*
1. Unadjusted analysis		
More than 1 BL ^1^	2.514 (1.079–5.858)	0.033
2. Adjusted analysis		
Cold ischemic time (minutes)	1.001 (0.999–1.002)	0.372
Donor age (years)	1.038 (1.004–1.073)	0.026
More than 1 BL ^1^	1.169 (0.941–1.452)	0.159

^1^ B-lines.

**Table 3 medicina-59-01931-t003:** Relationship between clinical data 3 days after kidney transplantation and kidney function.

	IGF ^1^	SGF+DGF ^1^	
**Recipients**	n = 51	n = 47	*p*
**mGFR (mL/min/1.73 m^2^)**	54.0 (29.0–71.0)	16.0 (8.0–34.5)	0.001
**Serum Sodium (mmol/L)**	137.0 (134.0–139.0)	136.0 (133.0–139.0)	0.125
**Serum albumin (g/L)**	31.0 (30.0–33.38)	31.0 (30.0–34.0)	0.664
**Hemoglobin (g/L)**	94.79 (16.37)	92.74 (12.78)	0.592
**CRP (mg/L)**	17.2 (7.5–26.0)	27.0 (11.9–46.56)	0.013
**Sum of BL ^2^**	4.48 (5.6)	5.16 (4.96)	0.54
**ECW/TBW ratio ^3^**	0.394 (0.385–0.404)	0.395 (0.387–0.404)	0.538
**Diuresis per 24 h (mL/24 h)**	3622.45 (1623.53)	2475.39 (1534.92)	0.001
**Fluid therapy per 24 h (mL/kg/24 h)**	46.86 (17.68)	39.55 (17.42)	0.068

^1^ Determination of IGF, SGF and DGF was performed using the following formula: the difference between serum creatinine (Scr) at 0 h and Scr on day 7 divided by Scr at 0 h. In those who did not require dialysis, SGF was defined as a creatinine reduction ratio less than 0.7 and IGF was defined as a ratio greater than or equal to 0.7. DGF was defined by at least one dialysis session within 7 days of transplant. ^2^ Sum of B-lines. ^3^ ECW/TBW ratio—extracellular water/total body water ratio. Data given as a number (%), mean (SD) or median (IQR).

**Table 4 medicina-59-01931-t004:** Relationship between clinical data 7 days after kidney transplantation and kidney function.

	IGF ^1^	SGF + DGF ^1^	
**Recipients**	n = 51	n = 47	*p*
**mGFR (mL/min/1.73 m^2^)**	64.94 (29.66)	30.38 (20.48)	0.001
**Serum Sodium (mmol/L)**	136.0 (135.0–138.0)	136.0 (135.0–138.0)	0.72
**Serum albumin (g/L)**	33.0 (31.0–35.0)	33.0 (29.5–36.0)	0.646
**Hemoglobin (g/L)**	102.08 (12.48)	96.62 (13.69)	0.042
**CRP (mg/L)**	5.0 (2.03–5.0)	6.35 (3.88–16.43)	0.002
**Sum of BL ^2^**	3.2 (3.39)	4.83(4.55)	0.064
**ECW/TBW ratio ^3^**	0.396 (0.390–0.406)	0.396 (0.390–0.404)	0.503
**Diuresis per 24 h (mL/24 h)**	3622.45 (1623.53)	2475.06 (1534.92)	0.001
**Fluid therapy per 24 h (mL/kg/24 h)**	54.68 (18.13)	38.83 (34.58)	0.001

^1^ Determination of IGF, SGF and DGF was performed using the following formula: the difference between serum creatinine (Scr) at 0 h and Scr on day 7 divided by Scr at 0 h. In those who did not require dialysis, SGF was defined as a creatinine reduction ratio less than 0.7 and IGF was defined as a ratio greater than or equal to 0.7. DGF was defined by at least one dialysis session within 7 days of transplant. ^2^ Sum of B-lines. ^3^ ECW/TBW ratio—extracellular water/total body water ratio. Data given as a number (%), mean (SD) or median (IQR).

**Table 5 medicina-59-01931-t005:** Relationship between clinical data on the day of discharge and kidney function.

	IGF ^1^	SGF+DGF ^1^	
**Recipients**	n = 51	n = 47	*p*
**mGFR (mL/min/1.73 m^2^)**	64.5 (24.23)	41.57 (17.79)	0.001
**Serum Sodium (mmol/L)**	136.0 (134.0–137.25)	137.0 (135.0–138.0)	0.092
**Serum albumin (g/L)**	34.3 (31.0–37.0)	34.4 (32.2–37.0)	0.482
**Hemoglobin (g/L)**	112.84 (13.77)	102.49 (13.50)	0.001
**CRP (mg/L)**	5 (1.19–5.0)	4.7 (1.0–5.0)	0.919
**Sum of BL ^2^**	1.34 (2.71)	1.20 (2.62)	0.823
**ECW/TBW ratio ^3^**	0.396 (0.390–0.406)	0.396 (0.393–0.407)	0.973
**Diuresis per 24 h (mL/24 h)**	3479.41 (963.8)	3066.3 (1093.72)	0.051
**Fluid therapy per 24 h (mL/kg/24 h)**	44.5 (12.63)	40.87 (13.88)	0.261

^1^ Determination of IGF, SGF and DGF was performed using the following formula: the difference between serum creatinine (Scr) at 0 h and Scr on day 7 divided by Scr at 0 h. In those who did not require dialysis, SGF was defined as a creatinine reduction ratio less than 0.7 and IGF was defined as a ratio greater than or equal to 0.7. DGF was defined by at least one dialysis session within 7 days of transplant. ^2^ Sum of B-lines. ^3^ ECW/TBW ratio—extracellular water/total body water ratio. Data given as a number (%), mean (SD) or median (IQR).

**Table 6 medicina-59-01931-t006:** Multivariate logistic regression for evaluation of factors relevant to worse kidney graft function (eGFR <30 mL/min/1.73 m^2^) after 1 year.

Model	Odds Ratio (95% CI)	*p*
Adjusted analysis		
Cold ischemic time (minutes)	1.000 (0.998–1.002)	0.848
Donor age (years)	0.936 (0.885–0.990)	0.021
More than 1 BL ^1^	1.090 (0.279–4.264)	0.902

^1^ B-lines.

## Data Availability

The data presented in this study are available on request from the corresponding author. The data are not publicly available due to limited ethical approval.

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
