# Peer review of "The Role of Pre- and Post-Transplant Hydration Status in Kidney Graft Recovery and One-Year Function"

_medicina, 2023, doi:10.3390/medicina59111931_

Round 1

Reviewer 1 Report

Comments and Suggestions for Authors

In the study presented, Bura A et al investigated the relationship between hydration status before and after transplantation and graft function in the first year after kidney transplantation. This was a single-center, prospective, observational study that included 98 kidney transplant recipients between September 2016 and January 2023. Bioimpedance analysis and lung ultrasonography (LU) were performed to assess hydration status and compare between patients with immediate graft function (IGF) and patients with slow and delayed graft function (SGF/DGF). The study found that lung congestion (higher sum of B-lines) detected by LU before transplantation was associated with slower graft recovery and worse kidney function at 1 year, whereas hyperhydration status detected by bioelectrical impedance analysis did not correlate with kidney graft function after transplantation.

The manuscript is of potential interest to the transplant community. Nevertheless, there are some issues and important limitations related to the reported research that need to be further addressed and clarified.

Major comments

1. Introduction

In the introduction, the authors should briefly present the main study results of previous studies examining hydration status (including studies using bioimpedance analysis) in relation to kidney transplantation outcomes.

2. Definitions of early graft function/dysfunction

The authors should also explain the definition of primary graft nonfunction and indicate whether these patients were included in the analysis.

3. Study population

a. The authors should explain why patients with early posttransplant complications (eg, surgical complications, renal artery/vein thrombosis, graft transplantectomy) were excluded from the analysis. This is a very interesting group to investigate the possible association between hydration status before and early after transplantation and worse outcomes after transplantation.

b. According to the exclusion criteria, the study cohort included 98 patients (of the 120 patients transplanted during the observation period, 22 were excluded). However, in the »Study Population" paragraph, the authors state that 25 recipients were excluded because of data limitations. This should be revised or clarified.

4. Determining the volume status

a. In patients who required hemodialysis immediately before transplantation, was volume status determined before or after the dialysis session?

b. I miss the body weight data before and just after transplantation. Did the authors record and track patient body weight at the same time that they performed LU and bioimpedance analysis? I would be interested in the correlations between the (dry) body weight and LU and the bioimpedance analysis before and after transplantation and posttransplant outcomes.

c. Previous studies have shown that hyperhydration status in the early posttransplant period is associated with increased abdominal pressure and posttransplant graft dysfunction. This may be reflected in the point- of-care intra-graft vein ultrasound waveform. Therefore, did the authors perform kidney graft Doppler/ultrasound after transplantation (eg, at days 3, 7, and at discharge)? This would be very interesting because the graft vein Doppler waveform becomes biphasic and monophasic (rather than continuous) in moderate and severe hyperhydration/congestion and this could be related to early graft (dys)function.

5. Baseline patient, donor and transplant-related characteristics and transplant outcomes

As shown in Table 1, patients were transplanted with DGF/SGF from older donors and after a prolonged cold ischemia time. Therefore, it is not surprising that these kidneys do not perform as well as kidneys transplanted from younger donors and with shorter cold ischemia. I wonder if LU before transplantation is independently associated with a higher probability of SGF/DGF when adjusting for other covariates in a multivariate regression model?

6. Post-operative fluid balance/volume status and fluids prescription

a. In addition to the fluid volume prescribed per day, data on body weight (as mentioned above) and body weight trends are lacking to additionally assess early postoperative fluid balance and hydration status.

b. The authors should explain the prescription of fluid therapy in patients with IGF vs DGF/SGF. What type of fluid did they use: isotonic or hypotonic saline, other? It seems that fluid therapy was not adjusted based on early graft function and hydration status assessment. From the data presented, it appears that all patients were hyperhydrated early after transplantation. What do the authors have to say about this problem? In my opinion, one of the most important goals of using additional tools (e.g., LU, bioimpedance) to assess the patient's volume status and prevent hyperhydration is to adjust fluid therapy prescription.

c. In my opinion, Tables 2, 3, and 4 can be combined into a single table.

7. Volume status and 1-year graft function

a. The authors should perform multivariate analysis to assess an independent effect of pretransplant hydration status and 1-year graft function. Different baseline characteristics (not only related to hydration status) could account for a different baseline risk of developing SGF/DGF but also worse graft function at 1 year.

b. In addition to assessing the association with hydration status before transplantation, the authors should also assess the association between hydration status early after transplantation (at days 3, 7, and discharge) and 1-year graft function.

8. Discussion and Study conclusions

The authors stated that this study demonstrates a new opportunity to use LU before kidney transplantation to predict early function of the transplanted kidney and function after 1 year and to intervene early. However, the study presented only examined the association between pretransplant hydration status and early graft function and was not a diagnostic study to assess the diagnostic value of LU /bioimpedance analysis. It also did not examine potential interventions (eg, fluid management). Therefore, the authors should revise the relevant sections in the Discussion/Conclusion sections of the manuscript.

Other comments

1. The English language is suboptimal, and I suggest that the manuscript be subjected to professional English language editing.

2. I recommend using the term “deceased donor” instead of “cadaveric donor”."

3. Results (page 4), postoperative data. What does “CRB level” mean? Is this a typo and should be written “CRP level”?

Comments on the Quality of English Language

The English language is suboptimal, and I suggest that the manuscript be subjected to professional English language editing.

Author Response

Response to Reviewer 1:

Dear Reviewer,

We appreciate your feedback. We corrected an article according to your suggestions. Thank you.

  1. Introduction

In the introduction, the authors should briefly present the main study results of previous studies examining hydration status (including studies using bioimpedance analysis) in relation to kidney transplantation outcomes.        Answer: In the introduction, we briefly presented the results of a previous study on hydration status in relation to kidney transplantation outcomes. This can be found in lines 64 to 74.

  1. Definitions of early graft function/dysfunction

The authors should also explain the definition of primary graft nonfunction and indicate whether these patients were included in the analysis.                   Answer: The definition of primary graft nonfunction was added in lines 96-97. These patients were excluded from the study.

  1. Study population:
    1. The authors should explain why patients with early posttransplant complications (eg, surgical complications, renal artery/vein thrombosis, graft transplantectomy) were excluded from the analysis. This is a very interesting group to investigate the possible association between hydration status before and early after transplantation and worse outcomes after transplantation.                                            Answer: We did not include recipients with early post-transplant complications and PNF due to the very small numbers: there were no possibilities to make a correct statistical analysis out of a few patients.
    2. According to the exclusion criteria, the study cohort included 98 patients (of the 120 patients transplanted during the observation period, 22 were excluded). However, in the »Study Population" paragraph, the authors state that 25 recipients were excluded because of data limitations. This should be revised or clarified.  Answer: In the early post-transplant period, we observed 98 recipients after kTx. On follow-up 1 year after kTx, we tracked 74 recipients from 98. 24 was excluded due to data limitations. 25 was an error of typing, we corrected this in the article.
  1. Determining the volume status:
    1. In patients who required hemodialysis immediately before transplantation, was volume status determined before or after the dialysis session?                                                                            Answer: BIA and LUS were performed approximately 2 hours before kTx. If the patient underwent HD, the BIA was performed after the HD session. Explanation is included in the article (lines 131 – 133).
    2. I miss the body weight data before and just after transplantation. Did the authors record and track patient body weight at the same time that they performed LU and bioimpedance analysis? I would be interested in the correlations between the (dry) body weight and LU and the bioimpedance analysis before and after transplantation and posttransplant outcomes.                                                                Answer: Body weight data was not recorded; we will record it in future studies as we believe it is valuable data, thank you very much for this suggestion.
    3. Previous studies have shown that hyperhydration status in the early posttransplant period is associated with increased abdominal pressure and posttransplant graft dysfunction. This may be reflected in the point-of-care intra-graft vein ultrasound waveform. Therefore, did the authors perform kidney graft Doppler/ultrasound after transplantation (eg, at days 3, 7, and at discharge)? This would be very interesting because the graft vein Doppler waveform becomes biphasic and monophasic (rather than continuous) in moderate and severe hyperhydration/congestion and this could be related to early graft (dys)function.                                                                      Answer: Thank you for your insight regarding this parameter. However, we were unable to perform it due to a lack of skills and limitations of our ultrasonography machine.
  1. Baseline patient, donor and transplant-related characteristics and transplant outcomes. As shown in Table 1, patients were transplanted with DGF/SGF from older donors and after a prolonged cold ischemia time. Therefore, it is not surprising that these kidneys do not perform as well as kidneys transplanted from younger donors and with shorter cold ischemia. I wonder if LU before transplantation is independently associated with a higher probability of SGF/DGF when adjusting for other covariates in a multivariate regression model?                                                              Answer: In the unadjusted analysis, having more than 1 BL in LUS was associated with a higher risk of getting SGF or DGF after kTx (OR 2.514, CI (1.079-5.858)). The multivariate regression model was executed, but due to the small sample size, the results are not presented. In the unadjusted analysis, the following results were obtained: Cold ischemic time: OR 1.001, CI (0.999-1.002), p-value 0.372; Deceased donor age: OR 1.038, CI (1.004-1.073), p-value 0.026 BL: OR 1.169, CI (0.941 – 1.452), p-value 0.159 The model’s AIC was 122.47 and Nagelkerke R Square was 0.2. Without BL in the model, the AIC increased to 130.910 and Nagelkerke R Square decreased to 0.114. We hypothesize that with an increased cohort size, BL will become statistically significant in the multivariate regression model.
  1. Post-operative fluid balance/volume status and fluid prescription:
    1. In addition to the fluid volume prescribed per day, data on body weight (as mentioned above) and body weight trends are lacking to additionally assess early postoperative fluid balance and hydration status.                                                                                          Answer: Body weight data were not recorded; we will record it in future studies as we believe it is valuable data.
    2. The authors should explain the prescription of fluid therapy in patients with IGF vs DGF/SGF. What type of fluid did they use: isotonic or hypotonic saline, other? It seems that fluid therapy was not adjusted based on early graft function and hydration status assessment. From the data presented, it appears that all patients were hyperhydrated early after transplantation. What do the authors have to say about this problem? In my opinion, one of the most important goals of using additional tools (e.g., LU, bioimpedance) to assess the patient's volume status and prevent hyperhydration is to adjust fluid therapy prescription.                                                                      Answer: The prescription of fluid therapy after kidney transplantation was determined according to the local protocol, which included a basic volume load of 1 liter of isotonic solution and 1 liter of 5% glucose solution with additional compensation of extra losses. We believe this is the main reason why all recipients were overhydrated early after kTx. We included this explanation into the article (lines 115-118).
    3. In my opinion, Tables 2, 3, and 4 can be combined into a single table.   Answer: We believe that separate figures provide a clearer view of the situation so we would like to leave them not corrected.
  1. Volume status and 1-year graft function:
    1. The authors should perform multivariate analysis to assess an independent effect of pretransplant hydration status and 1-year graft function. Different baseline characteristics (not only related to hydration status) could account for a different baseline risk of developing SGF/DGF but also worse graft function at 1 year.          Answer: The multivariate analysis did not show an independent effect of pre-transplant hydration status on 1-year graft function, possibly due to our small population size.
    2. In addition to assessing the association with hydration status before transplantation, the authors should also assess the association between hydration status early after transplantation (at days 3, 7, and discharge) and 1-year graft function.                                          Answer: Post-transplant hydration status did not have an association with graft function. (lines 205-207)
  1. Discussion and Study conclusions.                                                              The authors stated that this study demonstrates a new opportunity to use LU before kidney transplantation to predict early function of the transplanted kidney and function after 1 year and to intervene early. However, the study presented only examined the association between pretransplant hydration status and early graft function and was not a diagnostic study to assess the diagnostic value of LU /bioimpedance analysis. It also did not examine potential interventions (eg, fluid management). Therefore, the authors should revise the relevant sections in the Discussion/Conclusion sections of the manuscript.                          Answer: We have reviewed and made changes to the Discussion and Conclusion sections according to your recommendations. (lines 348-354).

Other comments:

  1. The English language is suboptimal, and I suggest that the manuscript be subjected to professional English language editing.

Answer: The English language has been reviewed and corrections have been made.

  1. I recommend using the term “deceased donor” instead of “cadaveric donor”.

Answer: We have changed the terminology from “cadaveric donor” to “deceased donor” according to your recommendation.

  1. Results (page 4), postoperative data. What does “CRB level” mean? Is this a typo and should be written “CRP level”?

Answer: The abbreviation “CRB” means “CRP” in English; I have reviewed and corrected this in the article.

Thank you once again for your valuable feedback.

Reviewer 2 Report

Comments and Suggestions for Authors

In this article, Bura et al. have analysed the hydration status of patients pre- and post-kidney transplant and its association with graft function one year after transplant. The hydration status was measured by bioimpedance analysis and lung ultrasonography. The main conclusion of the research is that the presence of lung congestion (> 1 B-line) before transplantation is associated with a 2.5 times higher risk of slower or delayed graft function and worse kidney function one year after transplant, while bioimpedance analysis showed no correlation with kidney function.

The presented results are interesting and valuable and the limitations and future perspectives of the research are nicely stated. However, there are several technical problems with the manuscript that need to be addressed:

1. The terminology and use of abbreviations need to be unified in the abstract and throughout the manuscript. For instance, both the terms lung sonography and lung ultrasound are used throughout the manuscript. I suggest using the term lung ultrasonography. The use of abbreviations is inconsistent throughout the manuscript. Abbreviations have to be defined the first time the term appears in the text and then only abbreviations should be used in the further text (the abstract is considered a separate text). A few examples where this is incorrect: 

line 23 B-lines are mentioned first (and the hyphen is missing), but the abbreviation BL is defined after, in line 26, instead of in line 23.

line 20 has the full names bioimpedance analysis and lung sonography written, even though their abbreviations are defined in line 16

lines 104 and 105 hemodialysis is written even though its abbreviation is described prior in line 47

end-stage renal disease is mentioned in line 39 without defining the abbreviation, but the abbreviation is used in line 54

These are just a few examples and all the abbreviations should be checked and corrected if necessary.

2. References need to be checked and corrected throughout as they are out of numerical order and incorrect in some instances. For example, references 7,8,16,17 (line 54) appear before reference 6 (line 66). The reference in line 47 cites the wrong study. Some references are missing from the manuscript altogether - references 23 and 37 are not mentioned anywhere in the manuscript.

3. The subsection "POST HOC ANALYSIS OF RELATION BETWEEN HEART LEFT VENTRICULAR HYPERTROPHY AND OVERHYDRATION" - "heart" is extra and should be removed. Also, a reference needs to be added for the definition of left ventricular hypertrophy by the American Society of Echocardiography and/European Association of Cardiovascular Imaging. You can also add a statement to the abstract that there was no association between hydration status and left ventricular hypertrophy as it is also a part of your results.

4. Line 169 and Tables 1-4, is CRB meant to be CRP (C-reactive protein)?

5. Figures and tables. Figure 2 - "Sum of B lines" should be added to the y-axis, similar to what is done in Figure 3a. Also, it should be stated what the error bars represent in the figure caption. I am assuming it is the SD. Figure 3b is very confusing. It is not clear what the different colored dots represent and a legend should be made similar to Figure 2. I can assume that blue represents IGF and orange SGF+DGF, but that leaves the grey dots. If the dots do represent the means of BIA for IGF and SGF+DGF groups, then the numbers in Figure 3b do not correspond to the numbers in Tables 1-4. Please re-check the numbers in the Tables/Figures and correct them where they are incorrect. 

6. If possible, could you add images of lung ultrasonography for readers to better visualize the B-lines? One image with no B-lines and one with several B-lines should be sufficient. This can be added as supplementary material.

Author Response

Dear Reviewer,

We appreciate your feedback. Your insights are of great value for improving our article. Thank you.

  1. The terminology and use of abbreviations need to be unified in the abstract and throughout the manuscript. For instance, both the terms lung sonography and lung ultrasound are used throughout the manuscript. I suggest using the term lung ultrasonography. The use of abbreviations is inconsistent throughout the manuscript. Abbreviations have to be defined the first time the term appears in the text and then only abbreviations should be used in the further text (the abstract is considered a separate text). A few examples where this is incorrect:

line 23 B-lines are mentioned first (and the hyphen is missing), but the abbreviation BL is defined after, in line 26, instead of in line 23.

line 20 has the full names bioimpedance analysis and lung sonography written, even though their abbreviations are defined in line 16

lines 104 and 105 hemodialysis is written even though its abbreviation is described prior in line 47

end-stage renal disease is mentioned in line 39 without defining the abbreviation, but the abbreviation is used in line 54

These are just a few examples and all the abbreviations should be checked and corrected if necessary.

Answer: Thank you for your notes, we corrected all terminology and abbreviations in the article.

  1. References need to be checked and corrected throughout as they are out of numerical order and incorrect in some instances. For example, references 7,8,16,17 (line 54) appear before reference 6 (line 66). The reference in line 47 cites the wrong study. Some references are missing from the manuscript altogether - references 23 and 37 are not mentioned anywhere in the manuscript.

Answer: The references were reviewed, and some were deleted. We also corrected the numerical order.

  1. The subsection "POST HOC ANALYSIS OF RELATION BETWEEN HEART LEFT VENTRICULAR HYPERTROPHY AND OVERHYDRATION" - "heart" is extra and should be removed. Also, a reference needs to be added for the definition of left ventricular hypertrophy by the American Society of Echocardiography and/European Association of Cardiovascular Imaging. You can also add a statement to the abstract that there was no association between hydration status and left ventricular hypertrophy as it is also a part of your results.

Answer: We removed the word “heart” from the topic “POST HOC ANALYSIS OF RELATION BETWEEN LEFT VENTRICULAR HYPERTROPHY AND OVERHYDRATION” and added your suggested reference (line 145). Reference number 18. Thank you.

  1. Line 176 and Tables 1-4, is CRB meant to be CRP (C-reactive protein)?

Answer: The abbreviation “CRB” means “CRP” in English. We reviewed and corrected it in the article.

  1. Figures and tables. Figure 2 - "Sum of B lines" should be added to the y-axis, similar to what is done in Figure 3a. Also, it should be stated what the error bars represent in the figure caption. I am assuming it is the SD. Figure 3b is very confusing. It is not clear what the different colored dots represent and a legend should be made similar to Figure 2. I can assume that blue represents IGF and orange SGF+DGF, but that leaves the grey dots. If the dots do represent the means of BIA for IGF and SGF+DGF groups, then the numbers in Figure 3b do not correspond to the numbers in Tables 1-4. Please re-check the numbers in the Tables/Figures and correct them where they are incorrect.

Answer: We added the label “Sum of B lines” to the y-axis in Figure 2. In Figures 3a and 3b, we attempted to show all participants’ hydration status during the observation period. This also explaned in lines 200-202 of the text.

  1. If possible, could you add images of lung ultrasonography for readers to better visualize the B-lines? One image with no B-lines and one with several B-lines should be sufficient. This can be added as supplementary material.

Answer: We did not make pictures of lung sonography images of our own patients. We think that taking them from the internet would be unethical. Pictures of lung sonography can be found in reference number 23.

I hope these revisions adequately address your concerns. Thank you once again for your time and practical comments.

Round 2

Reviewer 1 Report

Comments and Suggestions for Authors

The authors had revised the manuscript up to a point. However, one important point that was not addressed is the following:

- different baseline characteristics (not only related to hydration status) could be responsible for a different baseline risk for SGF/DGF development and also for worse graft function at 1 year. Therefore, the authors should present the results of both univariate and multivariate analysis (even if they are not statistically significant). The explanation given by the authors (about the small number of patients included in the cohort) should then be discussed in the Discussion/Limitation sections of the manuscript.

Comments on the Quality of English Language

In my opinion, English could be further improved.

Author Response

Dear Reviewer,

We appreciate your feedback. We corrected an article according to your suggestions. Thank you a lot for your help in improving the manuscript.

Comments and Suggestions for Authors:  different baseline characteristics (not only related to hydration status) could be responsible for a different baseline risk for SGF/DGF development and also for worse graft function at 1 year. Therefore, the authors should present the results of both univariate and multivariate analysis (even if they are not statistically significant). The explanation given by the authors (about the small number of patients included in the cohort) should then be discussed in the Discussion/Limitation sections of the manuscript.

Response: Dear Reviewer, we have incorporated multivariate analysis in our study, which can be found on lines 167-176, Table 2; and 231-238, Table 6. We have also discussed the limitations of our study pertaining to the small cohort size and suggested that an increased study population could yield more robust results. This discussion can be found on lines 364-374.

Comments on the Quality of English Language: In my opinion, English could be further improved.   Response: The English language has been corrected by MDPI and AUTHORsevices. Certificate attached
